∂ | **Open Peer Review** | Veterinary Microbiology | Research Article

# Integrative RNA-seq and ChIP-seq analysis unveils metabolic regulation as a conserved antiviral mechanism of chicken p53

Lu Cui,[1] Xuefeng Li,[1] Zhijie Chen,[1] Zheyi Liu,[1] Yu Zhang,[1] Zongxi Han,[1] Shengwang Liu,[1] Hai Li[1]

**ABSTRACT** The tumor suppressor p53, primarily functioning as a transcription factor, has exhibited antiviral capabilities against various viruses in chickens, including infectious bursal disease virus (IBDV), avian leukosis virus subgroup J (ALV-J), and avian infectious laryngotracheitis virus (ILTV). Nevertheless, the existence of a universal antiviral mechanism employed by chicken p53 (chp53) against these viruses remains uncertain. This study conducted a comprehensive comparison of molecular networks involved in chp53's antiviral function against IBDV, ALV-J, and ILTV. This was achieved through an integrated analysis of ChIP-seq data, examining chp53's genome-wide chromatin occupancy, and RNA-seq data from chicken cells infected with these viruses. The consistent observation of chp53 target gene enrichment in metabolic pathways, confirmed via ChIP-qPCR, suggests a ubiquitous regulation of host cellular metabolism by chp53 across different viruses. Further genome binding motif conservation analysis and transcriptional co-factor prediction suggest conserved transcriptional regulation mechanism by which chp53 regulates host cellular metabolism during viral infection. These findings offer novel insights into the antiviral role of chp53 and propose that targeting the virus-host metabolic interaction through regulating p53 could serve as a universal strategy for antiviral therapies in chickens.

**IMPORTANCE** The current study conducted a comprehensive analysis, comparing molecular networks underlying chp53's antiviral role against infectious bursal disease virus (IBDV), avian leukosis virus subgroup J (ALV-J), and avian infectious laryngotracheitis virus (ILTV). This was achieved through a combined assessment of ChIP-seq and RNA-seq data obtained from infected chicken cells. Notably, enrichment of chp53 target genes in metabolic pathways was consistently observed across viral infections, indicating a universal role of chp53 in regulating cellular metabolism during diverse viral infections. These findings offer novel insights into the antiviral capabilities of chicken p53, laying a foundation for the potential development of broad-spectrum antiviral therapies in chickens.

**KEYWORDS** chicken, p53, metabolism, ChIP-seq, RNA-seq

The tumor suppressor p53, often referred to as 'the guardian of the genome', operates as a transcription factor overseeing a multifaceted signal transduction network recognized as the p53 pathway (1–4). Within the extensive array of biological processes regulated by the p53 network, p53 demonstrates control over various aspects of host-immune responses, including innate immunity, nitric oxide-induced apoptosis, and histocompatibility complex I expression (5–8). These mechanisms stand as crucial defenses employed by the host to combat viral infections. Across numerous viruses, p53's role in antiviral function has been well-established, spanning vesicular stomatitis virus (9, 10), poliovirus (11), hepatitis C virus (12), influenza A virus (13), and Japanese encephalitis virus (14). For instance, p53 plays an essential role in enhancing the type I IFN-mediated

Address correspondence to Hai Li, lihai@xjtu.edu.cn, or Shengwang Liu, liushengwang@caas.cn.

Lu Cui and Xuefeng Li contributed equally to this article. The order of authors was determined alphabetically.

The authors declare no conflict of interest.

See the funding table on p. 15.

immune response against influenza A virus (IAV) infection. Silencing p53 expression via RNAi resulted in enhanced IAV replication, correlating with diminished expression of antiviral IFN-stimulated genes (15). Moreover, exogenous expression of p53 actively suppressed Japanese encephalitis virus (JEV) replication. When p53-deficient mice were infected with JEV, they exhibited significantly increased mortality rates and viremia compared to wild-type p53 mice (14). Despite more than 20 years of research into the antiviral role of p53, a comprehensive delineation of its specific antiviral mechanisms against distinct viruses remains an ongoing endeavor, yet to be fully consolidated and explored.

Infectious diseases pose a substantial threat to both humans and animals. Animals play a critical role as vectors or hosts in the transmission of diseases, with over 60% of infectious diseases in humans being zoonotic in nature (16, 17). Consequently, the effective management of infectious diseases in animals is not only vital for animal productivity but also imperative for safeguarding human health and promoting overall well-being. Avian leukosis virus (ALV), infectious bursal disease virus (IBDV), and avian infectious laryngotracheitis virus (ILTV) are three important pathogens that threaten the poultry industry. Notably, p53, recognized as an antiviral molecule, plays a pivotal role in combating all three pathogens. In chickens, the crucial involvement of p53 in ALV subgroup J (ALV-J) infection is evident through mutations linked to abnormal expression of chicken p53 (chp53) in ALV-J-associated myelocytomas (18) as well as the heightened ALV-J replication observed upon p53 knockout (19). The same phenomenon is observed in chickens infected with IBDV. The overexpression of chp53 inhibited IBDV replication, whereas chp53 inhibition led to the opposite effect (20). Our previous study also provided experimental evidence for the antiviral effect of chp53, which identified chp53 as the key determinant of ILTV infection protecting uninfected standby cells from ILTV-induced paracrine-regulated apoptosis, which reduces pathological damage and delays viral transmission (21). Despite unchanged activity of chp53 in ILTV infected cells, manipulating chp53 activity affects ILTV replication effectively in infected cells (21). Yet, the existence of a conserved antiviral mechanism governed by chp53 remains elusive. Unraveling such a mechanism could pave the way for the development of a universal strategy for antiviral therapies in chickens.

In this study, our aim was to comprehensively explore the universal antiviral mechanism orchestrated by chp53. To achieve this, we conducted an integrated analysis, comparing the molecular networks involved in chp53's antiviral function against IBDV, ALV-J, and ILTV. This analysis involved a combination of ChIP-seq data of the genome-wide chromatin occupancy of chp53 and RNA-seq data from chicken cells infected with the viruses mentioned above. The common enrichment of chp53 target genes in metabolic pathways was observed and validated by ChIP-qPCR, suggesting a conserved regulation of host cellular metabolism by p53 upon viral infection in chickens.

## MATERIALS AND METHODS

### Viral strain and cell culture

The ILTV strain ILTV-LJS09 was cultured in leghorn male hepatoma (LMH, ATCC CRL-2117) cells using Dulbecco's modified Eagle's medium (DMEM) supplemented with 10% fetal bovine serum (FBS), 100 units/mL penicillin, 100 g/mL streptomycin, and 2 mM L-glutamine, according to previously established methods (22). A multiplicity of infection (MOI) of 0.1 was used for ILTV infection in the transcriptome data, and samples were collected for high-throughput sequencing 36 hours post-infection (hpi). These cells were maintained at 37°C with 5% $CO_2$. For chromatin immunoprecipitation followed by qPCR (ChIP-qPCR) assay, LMH cells were seeded in 100 mm dishes 12 h before transfection and then harvested 24 h after transfection.

## Plasmids and transfection

Chicken p53 cDNA was integrated into the p3xFLAG-CMV-7.1 expression vectors, resulting in the generation of the Flag-chp53 recombinant plasmid. These plasmids were generously provided by Professors Zhiyong Ma and Yafeng Qiu from the Shanghai Veterinary Research Institute, Chinese Academy of Agricultural Science in Shanghai, China. LMH cells were seeded onto tissue culture plates 12 h prior to transfection. Transfection was conducted using Turbofect transfection reagent (R0531, Thermo Scientific, Rockford, IL) as per the manufacturer's instructions, with both plasmids administered at a dosage of 1 µg per $1.0 \times 10^6$ cells.

## Chromatin immunoprecipitation assays

Chromatin immunoprecipitation (ChIP) experiments followed a previously described protocol (23) with slight adjustments. Crosslinking of $5 \times 10^6$ LMH cells was performed in 1% formaldehyde for 10 min, followed by quenching of the crosslinking with 0.125 M glycine at room temperature for 5 min. DNA fragments were obtained by sonication (6 mm; Cole Parmer, Chicago, IL), maintaining an ice bath throughout to achieve DNA sizes ranging from 200 to 500 bp. Each ChIP experiment employed sheared chromatin from LMH cells and 5 µg of either anti-flag antibody or isotype control IgG2b. Pulldown was performed using Protein A/G plus-agarose beads as per the manufacturer's instructions (sc-2003, Santa Cruz Biotechnology, Santa Cruz, CA). Immunoprecipitated DNA was purified using the QIAquick PCR Purification Kit (28106, QIAGEN, Valencia, CA). Subsequent ChIPqPCR was performed with Luna Universal qPCR Master Mix (M3003L, NEB, Ipswich, MA) on a Bio-Rad CFX96 instrument following standard procedures. Primer sequences are detailed in Table 1, and all samples were analyzed in triplicate.

## RNA sequencing and ChIP sequencing

LMH cells were subjected to comprehensive gene expression profiling and chromatin occupancy analysis using RNA and DNA deep sequencing performed by Annoroad Gene Technology Co., Ltd. (Beijing, China), following standard procedures. Each experiment, ChIP-seq and RNA-seq, was conducted with four biological replicates. For RNA-seq, RNA extraction from cells utilized the RNeasy Plus Mini Kit (74314, QIAGEN, Hilden, Germany) following the manufacturer's instructions. ChIP-seq involved the preparation of 10 ng ChIP DNA for HiSeq 2500 sequencing using the TruSeq ChIP Sample Prep Kit (IP-202–1012, Illumina, San Diego, CA), adhering to the manufacturer's guidelines. Libraries for RNA-Seq were prepared with the TruSeq RNA Sample Prep Kits v2 (Illumina) as per the manufacturer's protocol, beginning with RNA fragmentation. Sequencing generated paired-end reads, each with a length of 150 bp.

**TABLE 1** Primer sequences for ChIP-qPCR

| Gene | Primer direction | Sequence (5′ to 3′) |
|---|---|---|
| POLE2 | F | GGAAAACATCCCATGCCACCC |
| | R | GACTTCAGCTCCGCCTGCC |
| POLE3 | F | CTGCCCAACGCCGTCATCACC |
| | R | CCCCGCGCCGAAGATCTCTGA |
| POLE | F | CAGGTCCGTCCGCCCGTGAT |
| | R | CGCCCCGCAGCACCATTCCC |
| CTH | F | TGGGCAATGTTTTCACCCTC |
| | R | AGTGATTTCAAAGGCCACCA |
| ALDH7A1 | F | TGCCTGCTCTGAACACTC |
| | R | ACCAGTGCATGAAATACAAC |
| ACO2 | F | GCAGCCAAAGCCCCATCCCT |
| | R | TCTTTCCCCGGCCCTCAGAGC |
| PDE9A | F | TGGCTCTGCTAAATCTCTCCACT |
| | R | ACAACCAAGTCCAAGCACCGAA |

## RNA-seq data analysis

The RNA sequencing (RNA-seq) data underwent analysis using the Galaxy web-based tool (24). Gene ontology and pathway analysis were conducted via DAVID, utilizing a modified Fisher's exact test with an EASE score, setting the threshold at a $P$ value of < 0.05 (25).

## ChIP-seq data analysis

ChIP-seq data analysis utilized Galaxy (24), R (http://www.r-project.org/), and TBtools (26) for Venn diagram plotting. Predictions of putative chp53 direct target genes' transcription factors were made using oPOSSUM v 3.0 (27). This prediction relied on DNA sequences within ±100 bp intervals of identified ChIP-seq peaks, with parameters set as follows: conservation cutoff = 40%; matrix score threshold = 85%; results sorted by Fisher score; and upstream/downstream region = 5000/5000.

## Statistical analysis

SPSS software (SPSS for Windows version 13.0, SPSS Inc., Chicago, IL) conducted all statistical analyses. Results are presented as mean ± standard deviation (SD) from multiple experiments. Differences between two groups were assessed using a two-tailed unpaired Student's $t$-test. A $P$ value < 0.05 was considered statistically significant for all analyses.

## RESULTS

### Integrated analysis of ChIP-seq and transcriptome data reveals the biological processes regulated by chp53 upon ILTV infection

ILTV, also known as gallid alphaherpesvirus 1, is a member of the family *Herpesviridae* and the subfamily *Alphaherpesvirinae*. Despite widespread immunization, ILTV infection still result in avian infectious laryngotracheitis (AILT), which leads to significant financial losses for the global chicken industry (28). Our previous research revealed that ILTV can accelerate pathological damage and death in host cells by rapidly inducing apoptosis in uninfected host cells. Using genome-wide transcriptome analyses in combination with a set of functional studies, we found that this paracrine-regulated effect requires the repression of p53 activity in uninfected cells (21). Subsequently, we performed ChIP-seq of chp53 in LMH cells upon chp53 overexpression, which identified 15,108 chp53-bound genes (23). According to the analysis workflow depicted in Fig. 1, we performed cross-analysis of the 15,108 chp53-bound genes with existing RNA-seq data from uninfected apoptotic cells upon ILTV infection (APC+) (21) and cells infected with ALV-J (29) or IBDV (bursa of Fabricius [30], natural killer cells [31]). The resulting overlapping genes are designated as direct targets of chp53 regulation subsequent to viral infection. Notably, across all three pathogens, whether genes were upregulated or downregulated post-viral infection, more than half of these genes exhibited an intersection with the chp53-bound genes (Fig. 1, bottom). In the ILTV-APC transcriptome data, out of 359 upregulated genes, 266 were identified as direct chp53 target genes, while out of 258 downregulated genes, 202 were direct chp53 target genes. For the RNA-seq data of ALV-J infection, 968 out of the 1207 upregulated genes and 814 out of the 1089 downregulated were direct chp53 targets. In the IBDV-bursa of Fabricius RNA-seq data set, 1093 out of 1,507 upregulated genes were identified as direct chp53 targets, and among 576 downregulated genes, 432 were direct chp53 targets. Finally, in the IBDV-natural killer cells RNA-seq data set, 626 out of 1,004 upregulated genes were direct chp53 targets, while 720 out of 1,078 downregulated genes were direct chp53 targets.

To further investigate the biological functions exerted by these genes directly regulated by chp53 post-viral infection, we conducted an in-depth analysis of the molecular signaling pathways enriched by the upregulated and downregulated gene

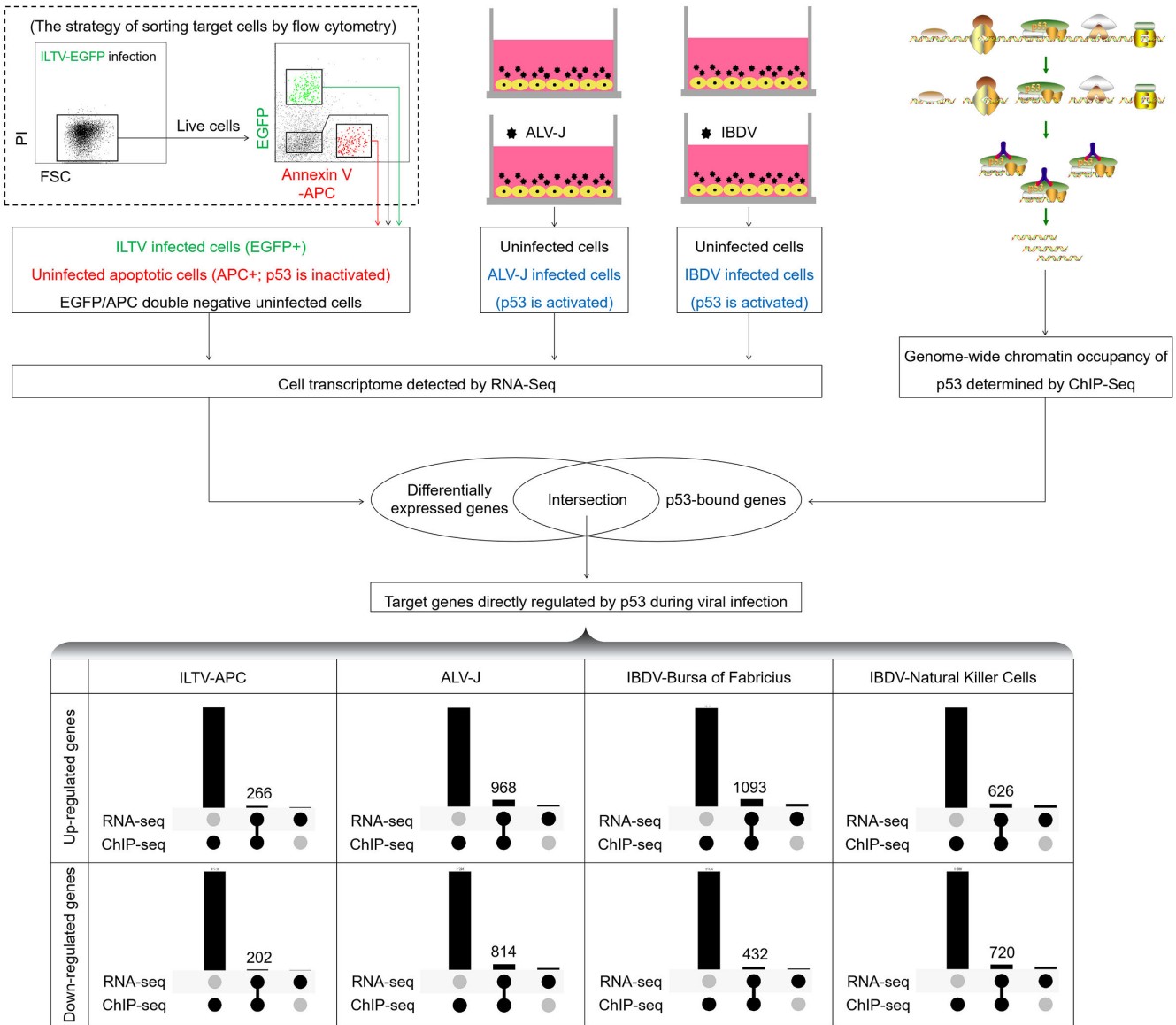

**FIG 1** Illustration of the experimental workflow of the integrated analysis of RNA-seq and ChIP-seq data. ILTV: infectious laryngotracheitis virus; ALV-J: avian leukosis virus subgroup J; IBDV: infectious bursal disease virus.

sets within each group. Our findings revealed that among the significantly upregulated genes in uninfected apoptotic cells upon ILTV infection, 266 target genes directly regulated by chp53 were primarily enriched in metabolic pathways such as carbon metabolism, biosynthesis of antibiotic, biosynthesis of amino acids, and glycolysis/gluconeogenesis (Fig. 2A). Conversely, within the significantly downregulated genes, 202 target genes directly regulated by chp53 were enriched in RNA transport, ribosome biogenesis in eukaryotes, spliceosome, protein processing in endoplasmic reticulum, and cell cycle pathways (Fig. 2B). These findings suggest unrestricted metabolic processes upon the inactivation of chp53 during ILTV infection.

## Integrated analysis of ChIP-seq and transcriptome data reveals the biological processes regulated by chp53 upon ALV-J infection

Avian leukemia, a prevalent tumor disease in birds, spreads both horizontally and vertically among chickens. Seven subgroups of avian leukemia virus are identified in

A

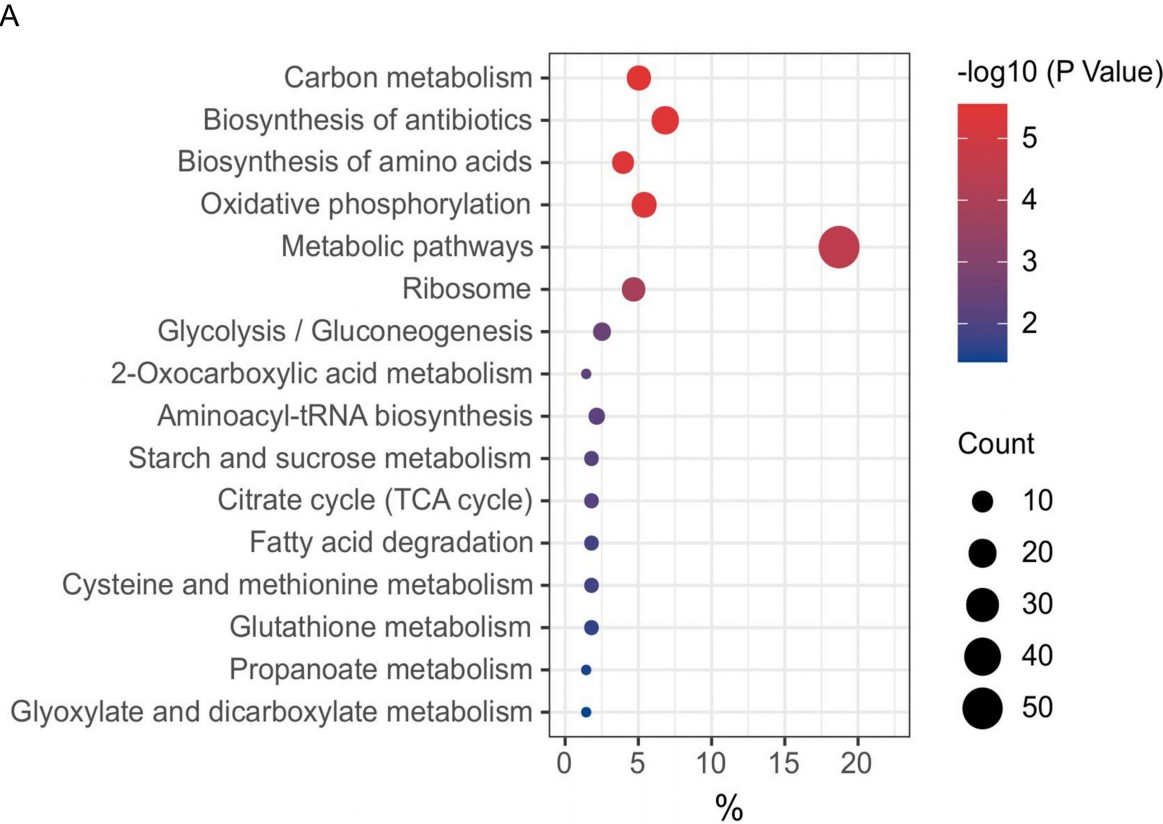

B

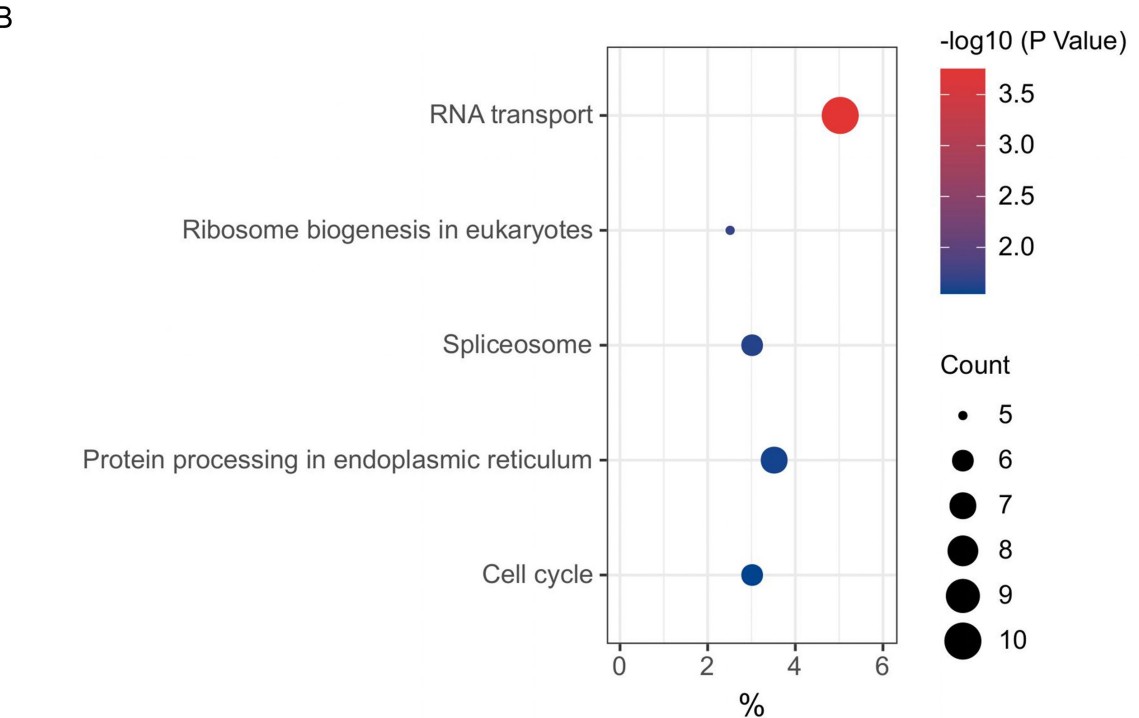

**FIG 2** KEGG enrichment analysis of the genes directly regulated by chp53 in uninfected apoptotic cells upon ILTV infection. (A) Enriched pathways of significantly upregulated genes. (B) Enriched pathways of significantly downregulated genes. The x-axis represents the gene ratio, that is, the number of differential genes per term/total differential genes, the y-axis represents the KEGG name of pathway, and the color represents the -log$_{10}$ (P value); the size of the dots represents the number of genes. TCA: tricarboxylic acid cycle.

chickens (A-E/J/K). Notably, avian leukosis virus subgroup J (ALV-J), a recombinant virus from endogenous and exogenous sources, significantly heightens pathogenicity and transmission compared to other subgroups (32, 33). ALV-J induces extensive tumor growth across multiple organs in infected chickens, leading to severe immunosuppression and high mortality rates. In the study of Zhang Hui et al., ALV-J infection activates chp53, shown by increased expression of key downstream target genes of p53 (*c-myc*, *p21*, *p27*, *bcl-2* and *bak*). This activation prevents ALV-J-induced inhibition of innate immune responses in DF-1 cells (chicken embryo fibroblast cell line). Additionally, their research demonstrates that chp53 inhibits ALV-J replication by upregulating the oncogene c-myc, causing cell cycle arrest at G1 and G2 phases, and promoting apoptosis in infected cells by regulating apoptosis-related genes *bcl-2* and *bak* (19).

Here, we identified novel chp53 target genes and their mechanisms of action by comparing ChIP-seq data with ALV-J infection-related transcriptome data obtained from GSE103207 (29). As illustrated in Fig. 3A, our analysis revealed that in ALV-J infected cells, among the 1207 significantly upregulated genes, the 968 genes directly regulated by chp53 were prominently enriched in signaling pathways such as MAPK signaling pathway, p53 signaling pathway, ubiquitin mediated proteolysis, and cytokine-cytokine receptor interaction. Among the 1089 significantly downregulated genes, the 814 genes directly regulated by chp53 were primarily enriched in signaling pathways associated with purine metabolism, oocyte meiosis, mismatch repair, pyrimidine metabolism, and others. The repression of pathways related to nucleotide metabolism upon chp53 activation by ALV-J infection is in consistent with the unrestricted metabolic processes upon the inactivation of chp53 caused by ILTV infection (Fig. 2A), which further demonstrates the metabolic regulation function of chp53 during viral infection.

## Integrated analysis of ChIP-seq and transcriptome data reveals the biological processes regulated by chp53 upon IBDV infection

Infectious bursal disease (IBD) is characterized by the immune suppression of infected birds. Wei Ouyang et al. established an *in vitro* infection model based on DF-1 cells to determine the antiviral effects of chp53 on IBDV infection. It was found that the expression level and activity of chp53 were remarkably increased in IBDV-infected DF-1 cells. The overexpression of chp53 inhibited IBDV replication and upregulated the expression of multiple chicken antiviral innate immunity genes (*IPS-1*, *IRF3*, *PKR*, *OAS*, and *Mx*), whereas the suppression of chp53 led to the opposite effect. This research indicates that chp53 activates the antiviral innate immune response against IBDV infection in chicken cells (20).

Figure 4 shows the data derived from the bursae of Fabricius infected with IBDV strain CJ801 (GSE94500) (30). There were a total of 1093 genes directly upregulated by chp53 upon IBDV infection, prominently enriched in signaling pathways such as cytokine-cytokine receptor interaction, Jak-STAT signaling pathway, and toll-like receptor signaling pathway (Fig. 4A). Conversely, the genes directly downregulated by chp53 were primarily enriched in signaling pathways related to purine metabolism, metabolic pathways, pyrimidine metabolism, base excision repair pathway, and others (Fig. 4B). Figure 5 show the data derived from the intraepithelial natural killer (IEL-NK) cells infected with the vvIBDV (very virulent IBDV) strain UPM0081 (GSE123920) (31). The genes directly upregulated by chp53 are primarily enriched in pathways such as influenza A and the p53 signaling pathway (Fig. 5A). Among the 24 pathways enriched by the genes directly downregulated by chp53, 19 are related to metabolic pathways (Fig. 5B).

## Comparative analysis of metabolic signal pathways

Through the integrated analysis of our ChIP-seq data in combination with the aforementioned RNA-seq data, we discovered that metabolic pathways are commonly regulated by chp53 in chicken cells upon viral infection. Within the uninfected apoptotic cells induced by ILTV infection, in which chp53 is inactivated, extensive promotion of metabolic pathways was observed. While in cells infected with ALV-J and IBDV, in which

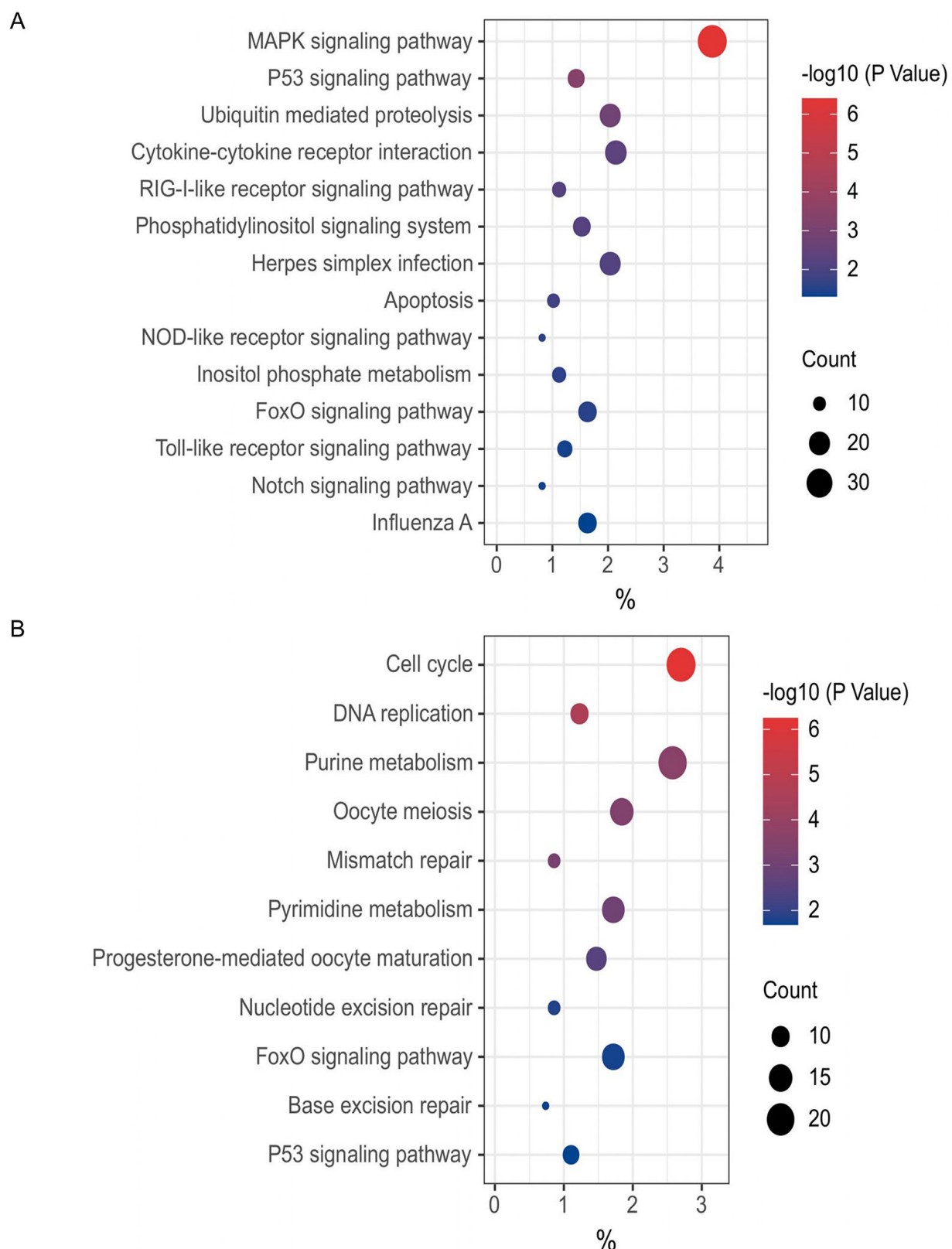

**FIG 3** KEGG enrichment analysis of the genes directly regulated by chp53 in cells infected with ALV-J. (A) Enriched pathways of significantly upregulated genes. (B) Enriched pathways of significantly downregulated genes. The *x*-axis represents the gene ratio, that is, the number of differential genes per term/total differential genes, the *y*-axis represents the KEGG name of pathway, and the color represents the $-\log_{10}$ (*P* value); the size of the dots represents the number of genes.

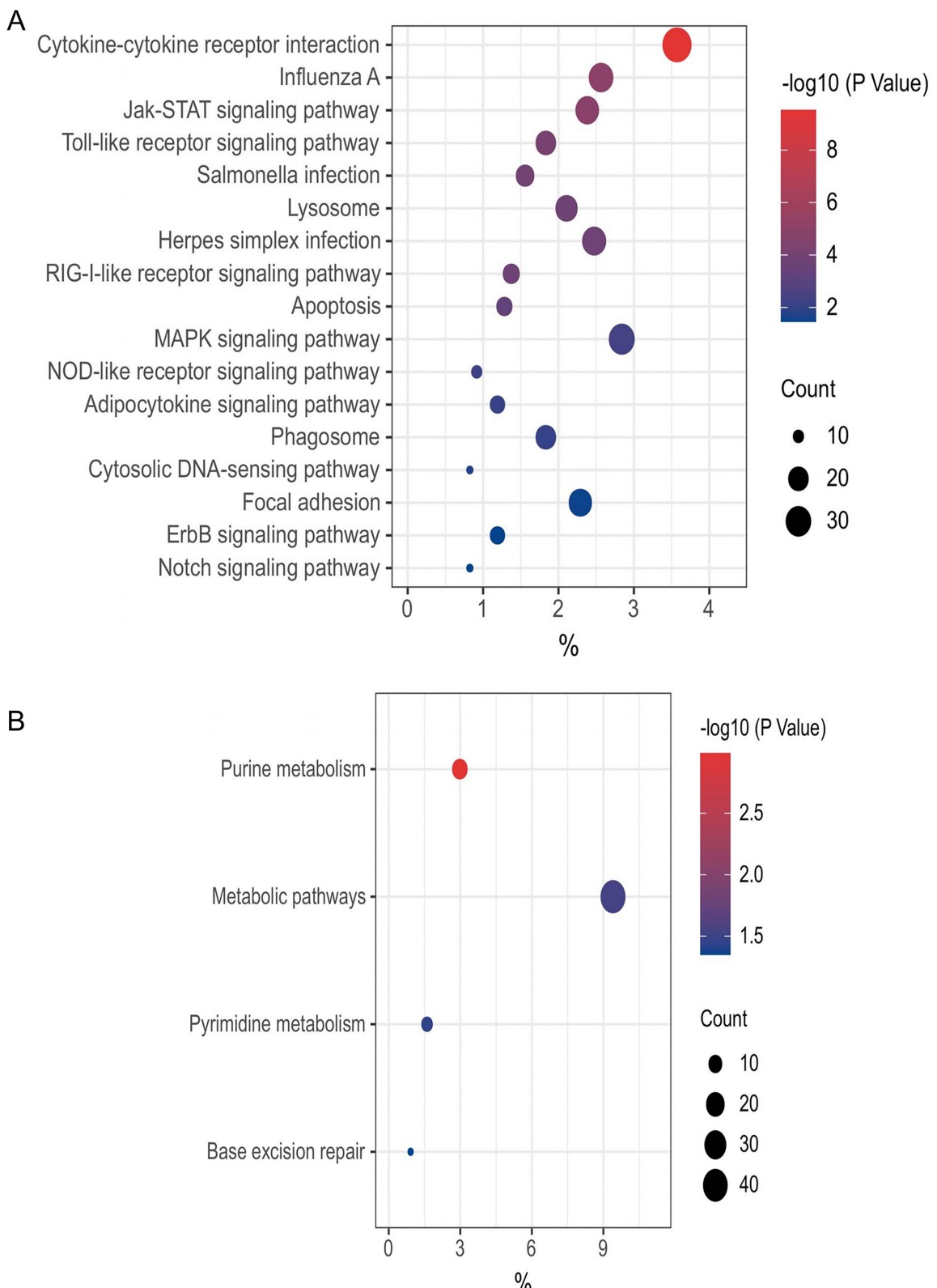

**FIG 4** KEGG enrichment analysis of the genes directly regulated by chp53 in bursa cells infected with IBDV. (A) Enriched pathways of significantly upregulated genes. (B) Enriched pathways of significantly downregulated genes. The *x*-axis represents the gene ratio, that is, the number of differential genes per term/total differential genes, the *y*-axis represents the KEGG name of pathway, and the color represents the -$\log_{10}$ (*P* value); the size of the dots represents the number of genes.

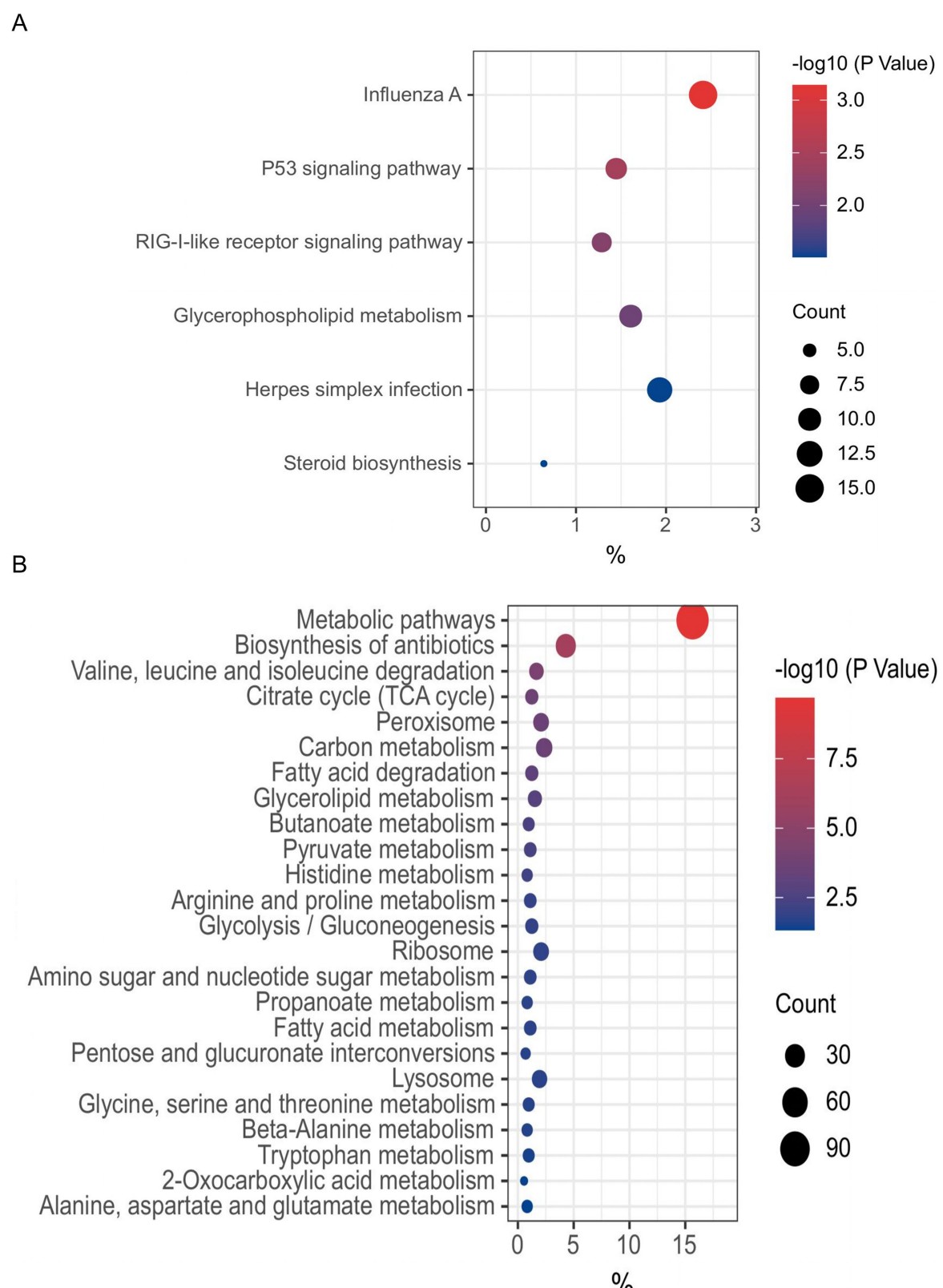

**FIG 5** KEGG enrichment analysis of the genes directly regulated by chp53 in IEL-NK cells infected with IBDV. (A) Enriched pathways of significantly upregulated genes. (B) Enriched pathways of significantly downregulated genes. The *x*-axis represents the gene ratio, that is, the number of differential genes per term/total differential genes, the *y*-axis represents the KEGG name of pathway, and the color represents the -log$_{10}$ (*P* value); the size of the dots represents the number of genes. TCA: tricarboxylic acid cycle.

chp53 is activated, the metabolic pathways were predominantly repressed. Further comparison of these metabolic pathways revealed the "metabolic pathways" is common affected among the four datasets, as depicted in Fig. 6. Pyrimidine and purine metabolism were commonly regulated by ALV-J and IBDV infection, suggesting that chp53 may exert antiviral effects by suppressing host nucleotide metabolic pathways upon the infections of these viruses. More broad metabolic processes, such as fatty acid degradation, glycolysis, gluconeogenesis, propanoate metabolism, tricarboxylic acid cycle (TCA cycle), antibiotic biosynthesis, carbon metabolism, and 2-oxocarboxylic acid metabolism, were commonly regulated by ILTV and IBDV infection.

Among the differently expressed chp53-bound genes involved in these common metabolic pathways shown in Fig. 6, nine genes were found to be conservatively regulated by chp53 upon ILTV, ALV-J, and IBDV infection: *PRIM2*, *POLE2*, *POLE3*, *POLE*, *CTH*, *ALDH7A1*, *ACO2*, and *PDE9A* (Fig. 7A). To confirm the direct transcriptional regulation of these common metabolic pathways by chp53 upon viral infection, we validated the presence of chp53 binding sites in the promoter regions of seven of these genes using ChIP-qPCR. The results were generally consistent with the ChIP-seq data (Fig. 7B), confirming these metabolism-related genes as conserved direct targets of chp53 upon viral infection.

## Chicken p53 regulates host cellular metabolism through conserved transcriptional regulation mechanism

p53 stands as a pivotal transcription factor, orchestrating a multitude of biological processes and pathways, including cell apoptosis, DNA repair, cell cycle regulation, and anti-tumor responses (34). Notably, it governs an array of target genes through an intricate transcriptional regulatory system, thereby modulating their transcriptional levels. However, the individual action of p53 often falls short in achieving such regulation and necessitates the involvement of specific auxiliary factors. These co-factors interact with p53, consequently affecting its binding ability to specific genes and regulating the activation or inhibition of cellular signaling pathways (35). To explore the transcriptional regulation mechanism by which chp53 regulates host cellular metabolism upon viral infection, DNA-binding motif conservation analysis was conducted using MEME program with the chp53-bound metabolic genes significantly regulated by viral infection. As depicted in Fig. 8A, a noteworthy resemblance between the binding motifs of chp53 in the upstream of these metabolic genes and the consensus p53

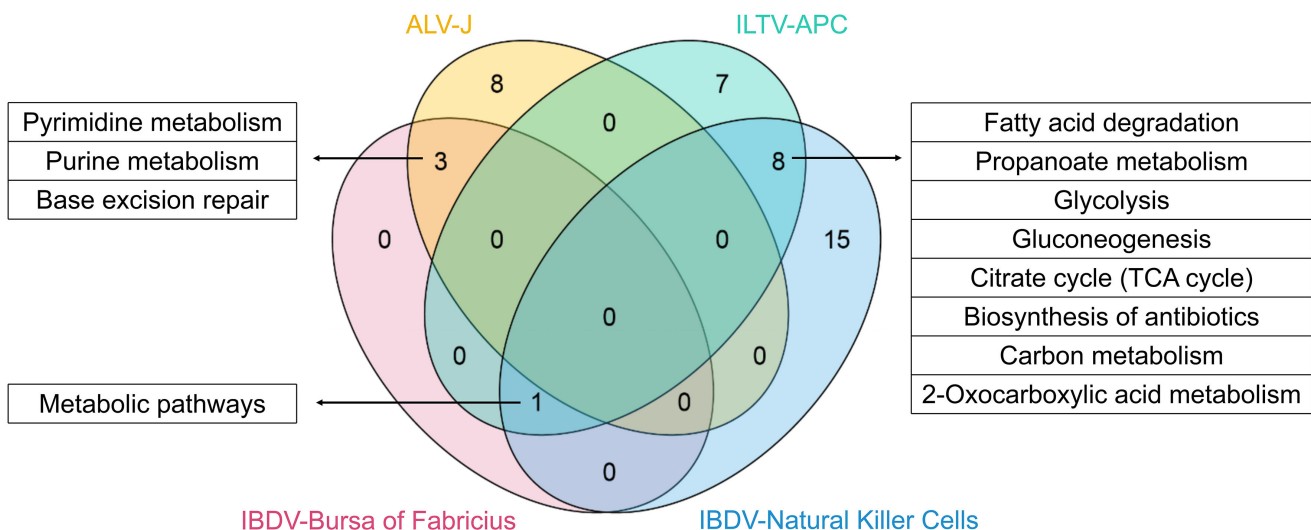

**FIG 6** Venn diagram of the metabolic pathways altered by viral infection. ALV-J: avian leukosis virus subgroup J; ILTV: infectious laryngotracheitis virus; TCA: tricarboxylic acid cycle.

A

| Pyrimidine metabolism | Purine metabolism | Fatty acid degradation | Glycolysis / Gluconeogenesis | Citrate cyele (TCA cyele) | Biosynthesis of antibiotics | Carbon metabolism | 2-Oxocarboxylic acid metabolism | Metabolic pathways |
|---|---|---|---|---|---|---|---|---|
| PRIM2 | PRIM2 | ALDH7A1 | ALDH7A1 | ACO2 | CTH | ACO2 | ACO2 | CTH |
| POLE2 | POLE2 | | | | ACO2 | | | |
| POLE3 | POLE3 | | | | ALDH7A1 | | | |
| POLE | POLE | | | | | | | |
| | PDE9A | | | | | | | |

B

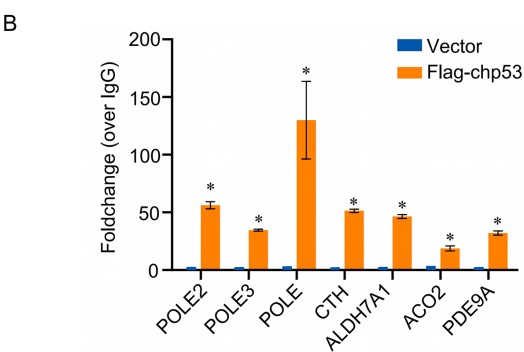

**FIG 7** ChIP-qPCR validation of metabolism-related genes conservatively regulated by chp53 during viral infection. (A) Summary of conserved genes in the common metabolic pathways presented in Fig. 6. (B) ChIP-qPCR validation of selected metabolism-related genes (*n* = 4). Asterisks indicate a significant difference (*, $P < 0.05$). TCA: tricarboxylic acid cycle.

binding motif was seen for all data sets, indicating the existence of a common transcriptional regulation mechanism by which chp53 regulates these metabolic pathways upon viral infection. To further explore the co-factors of chp53 in its regulation of cellular metabolism, we predicted the upstream regulatory factors of each group of metabolic genes, employing ChEA3-ChIP-X Enrichment Analysis Version 3. Consistent with the conserved chp53 binding motif in metabolic genes (Fig. 8A), the outcomes of analysis demonstrated remarkable consistency in the transcriptional factors (TFs) associated with these metabolic genes across the three pathogenic scenarios, since 63 out of the 102 predicted TFs were commonly shared by all groups (Fig. 8B). By sorting the betweenness values calculated via CytoNCA, the top five core transcriptional regulatory factors were identified, including Myc, CEBPA, CEBPB, SMAD3, and Fos (Fig. 8C). Research has shown that ALV infection can enhance the expression of the *c-myc* gene, thereby influencing cellular metabolism and contributing to tumor formation (36–38). Additionally, p53 has been found to inhibit the activity of c-myc (19), yet the mutual interaction between *c-myc* and the *bcl-2* gene has the capacity to inhibit p53 function (39). These reciprocal interactions suggest a complex regulatory relationship between p53 and c-myc. In the TGF-β signaling pathway, p53 forms a complex with SMAD proteins, leading to the activation of a cluster of genes regulating cellular processes such as cell proliferation, survival, apoptosis, and senescence. Although the molecular basis of this mutual dependency requires further elucidation, a subset of genes regulated by TGF-β1 possesses binding sequences for both p53 and SMAD proteins (40–43). Moreover, p53 not only directly activates the transcription of the CEBPA gene but the loss of its function could potentially lead to a decrease in CEBPA protein levels. Other studies have indicated that activation of the CEBPB/TRIM2/p53 signaling axis may promote tumor development. Notably, CEBPB modulates p53 activity in a cell cycle-dependent manner by regulating a regulatory site on the p53 promoter (44–46). The *Fos* gene also plays a crucial role by influencing the metabolic activity of host cells, thereby impacting the infection of ILTV (47). Additionally, we have discovered an interaction between Fos and p53 proteins, thereby regulating the transcription of the early ILTV gene, *ICP4* (48). These

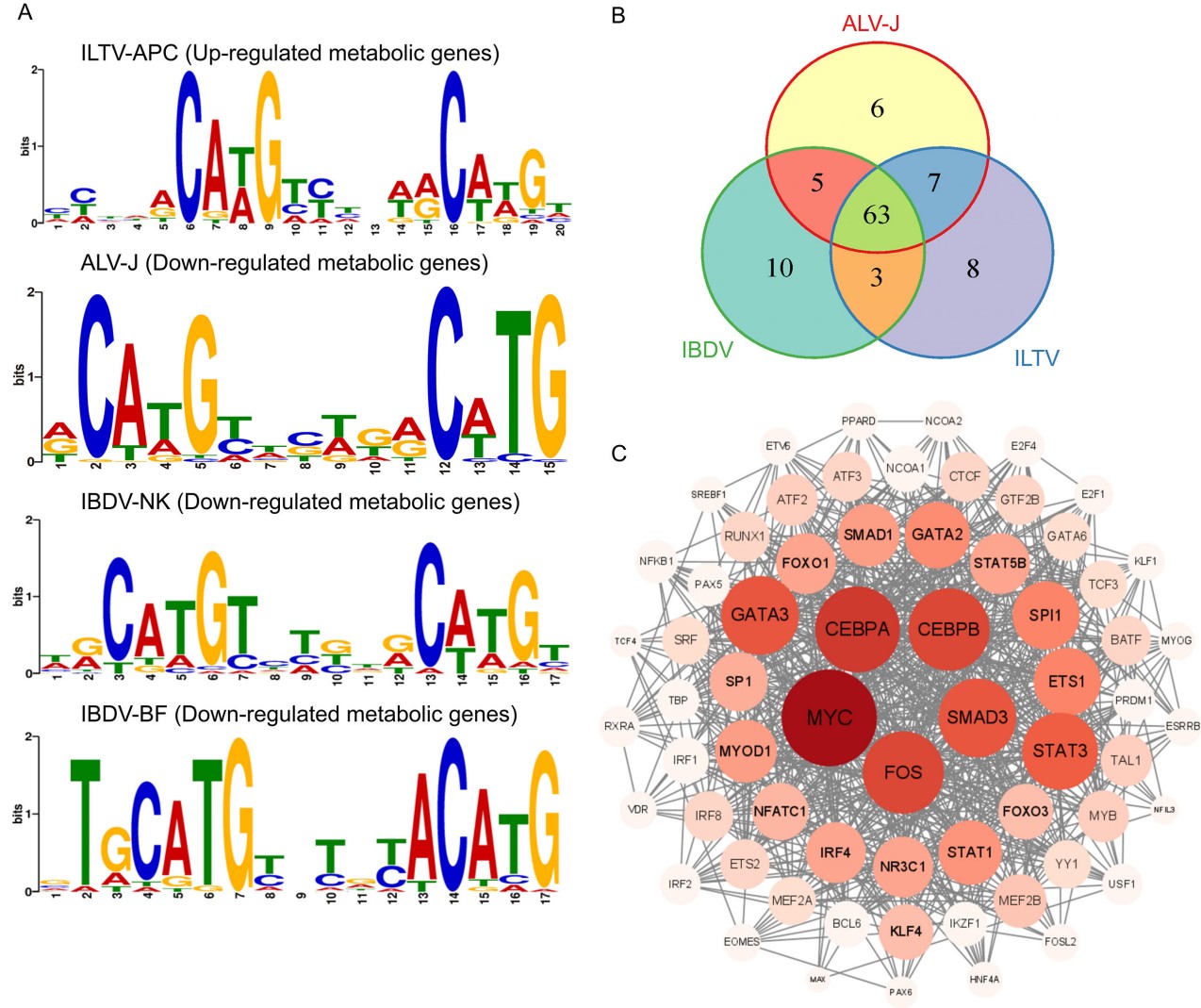

**FIG 8** Prediction of the binding motifs of chp53 in metabolic genes and its co-factors. (A) MEME algorithm was employed to assess the conservation of upstream chp53 binding motifs in the differently expressed chp53-bound metabolic genes. (B) Venn diagram demonstrates the overlap among predicted upstream transcriptional factors. (C) Analysis of the functional interactions among predicted transcription factors, with circle sizes determined by their betweenness values. ILTV: infectious laryngotracheitis virus; ALV-J: avian leukosis virus subgroup J; IBDV: infectious bursal disease virus.

findings further suggest that these transcription factors may act as co-factors in the transcriptional regulation of metabolism by chp53 during viral infection, supporting our analytical conclusions.

## DISCUSSION

To date, the antiviral functions of chp53 have been confirmed in many avian viruses, such as avian infectious laryngotracheitis virus (21), avian leukosis virus subgroup J (19), and infectious bursal disease virus (20). However, whether there is any universal antiviral mechanism of chp53 remains unclear. Here, to gain deeper insights into chp53 antiviral functions, we integrated ChIP-seq and RNA-seq analyses to comprehensively compare the molecular networks of chp53 against IBDV, ALV-J, and ILTV. The enrichment of chp53 target genes in metabolic pathways was commonly observed upon the infection of

different viruses and validated by ChIP-qPCR, suggesting a universal regulation of host cellular metabolism by p53 upon viral infection in chickens.

p53 exhibits diverse biological functions and regulates numerous target genes through a highly complex transcriptional regulatory mechanism. Thus, elucidating its target genes and co-factors is crucial for understanding and controlling its corresponding functions. At present, only a limited number of genes such as *p21* and *MDM2* have been identified as direct targets of chp53 (49). Moreover, our previous research has established three nucleotide metabolism genes (*RRM2*, *NME2*, *NME3*) and five genes responsible for ATP synthesis (*ATP5C1*, *NDUFA4*, *COX5A*, *NDUFC2*, and *NDUFB4*) as direct targets of chp53 (50). Given the diverse cellular origins of the high-throughput data utilized in the present study, conducting additional *in vivo* validations of chp53's direct metabolic influence during infections of a wider array of pathogens could aid in assessing the universality of our results.

This discovery significantly expanded our understanding of chicken p53's transcriptional regulation. Furthermore, we validated the involvement of chp53 in newly identified metabolic pathways through genes such as *PRIM2*, *POLE2*, *POLE3*, *POLE*, *CTH*, *ALDH7A1*, *ACO2*, and *PDE9A*. Additionally, we identified potential co-factors involved in chp53-mediated metabolic regulation. Considering that each step of viral replication is determined by the interaction between viruses and host cells, viral replication necessitates different nutritional requirements in infected cells compared to uninfected ones. To enhance their survival and replication chances, viruses evolve to dominate cellular processes, including metabolic pathways, to support their needs (51). Our analysis revealed concurrent enrichment of downstream p53-regulated target genes in metabolic pathways upon infection with various viruses. This suggests the transcriptional regulation of host cellular metabolism as a universal antiviral mechanism of p53 in chickens. These findings are consistent with our previous research of the essential metabolic requirement of ILTV replication and the apoptosis of bystander cell due to the metabolic abnormalities caused by the paracrine signaling induced by ILTV infection (50).

In terms of antiviral mechanisms, research has revealed that p53 plays a significant role by modulating cellular metabolic pathways (52). Specifically, studies have shown that p53 can influence the metabolic status of host cells, thereby impacting viral replication and transmission. p53 may suppress viral replication by regulating key metabolic pathways such as nucleotide metabolism, fatty acid synthesis, and glucose metabolism (52–55). Additionally, p53 may also modulate factors like cellular tryptophan and iron metabolism, thereby affecting the host cell's resistance to viruses (56, 57). Regulating metabolic pathways is considered one of the crucial strategies for treating viral infections (58). Conversely, viral infections alter the metabolic environment of host cells, making it more conducive to viral survival and replication. Therefore, by modulating metabolic pathways, especially those critical for viral requirements, it is possible to effectively inhibit viral growth and transmission. For instance, inhibiting pathways essential for viral nucleotide synthesis or glucose metabolism may reduce the rate of viral replication and mitigate the severity of viral infections (55, 59).

In conclusion, this study elucidates the widespread antiviral function of chp53 through the regulation of host cellular metabolic pathways. Furthermore, the modulation of metabolic pathways offers an important strategy for the treatment of viral infections. Therefore, targeting viral-host metabolic interactions through proper regulation of p53 may be a universal strategy for the antiviral therapy in chickens, offering insights for future development of broad-spectrum antiviral agents.

## ACKNOWLEDGMENTS

This work was supported by the Natural Science Foundation of Heilongjiang Province of China (grant number JQ2021C006), the National Natural Science Foundation of China (grant number 32072853), and the Agriculture Research System of China (grant number CARS-40-K18).

Conceptualization, H.L. and S.L.; validation, L.C., X.L., Z.C., Z.L., Y.Z. and Z.H.; formal analysis, L.C., X.L., Z.C., and Z.L.; investigation, L.C., X.L., Z.C., Z.L., Y.Z. and Z.H.; resources, H.L. and S.L.; writing-original draft preparation, L.C., X.L., Z.C., Z.L., and H.L.; writing-review and editing, L.C., X.L., Z.C., Z.L., H.L. and S.L.; supervision, H.L. and S.L.; funding acquisition, H.L. and S.L. All authors have read and agreed to the published version of the manuscript.

## AUTHOR AFFILIATION

[1]Division of Avian Infectious Diseases, State Key Laboratory for Animal Disease Control and Prevention, National Poultry Laboratory Animal Resource Center, Harbin Veterinary Research Institute, the Chinese Academy of Agricultural Sciences, Harbin, China

## AUTHOR ORCIDs

Lu Cui ⓘ http://orcid.org/0009-0006-3584-4934
Shengwang Liu ⓘ http://orcid.org/0000-0002-2854-8305
Hai Li ⓘ http://orcid.org/0000-0003-4772-2097

## FUNDING

| Funder | Grant(s) | Author(s) |
|---|---|---|
| Natural Science Foundation of Heilongjiang Province (Heilongjiang Natural Science Foundation) | JQ2021C006 | Hai Li |
| MOST \| National Natural Science Foundation of China (NSFC) | 32072853 | Hai Li |
| Agriculture Research System of China (China's Agricultural Research System) | CARS-40-K18 | Shengwang Liu |

## AUTHOR CONTRIBUTIONS

Lu Cui, Formal analysis, Investigation, Validation, Writing – original draft, Writing – review and editing | Xuefeng Li, Formal analysis, Investigation, Validation, Writing – original draft, Writing – review and editing | Zhijie Chen, Formal analysis, Investigation, Validation, Writing – original draft, Writing – review and editing | Zheyi Liu, Formal analysis, Investigation, Validation, Writing – original draft, Writing – review and editing | Yu Zhang, Investigation, Validation | Zongxi Han, Investigation, Validation | Shengwang Liu, Conceptualization, Funding acquisition, Resources, Supervision, Writing – review and editing | Hai Li, Conceptualization, Funding acquisition, Resources, Supervision, Writing – original draft, Writing – review and editing

## DATA AVAILABILITY

Raw data for both RNA-seq and ChIP-seq were deposited in the National Center for Biotechnology Information database under the accession numbers GSE193188 and GSE200405. Other data presented in this study are available on request from the corresponding author.

## ADDITIONAL FILES

The following material is available online.

### Open Peer Review

**PEER REVIEW HISTORY (review-history.pdf).** An accounting of the reviewer comments and feedback.

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
