## [Reviewer comments · Microbiology Spectrum]

Microbiology Spectrum

Integrative RNA-seq and ChIP-seq analysis unveils metabolic regulation as a conserved antiviral mechanism of chicken p53

Lu Cui, Xuefeng Li, Zhijie Chen, Zheyi Liu, Yu Zhang, Zongxi Han, Shengwang Liu, and Hai Li

Corresponding Author(s): Hai Li, Xi'an Jiaotong University

Review Timeline:

Submission Date:	February 2, 2024
Editorial Decision:	April 23, 2024
Revision Received:	April 26, 2024
Accepted:	May 2, 2024

Editor: Manjula Kalia

Reviewer(s): Disclosure of reviewer identity is with reference to reviewer comments included in decision letter(s). The following individuals involved in review of your submission have agreed to reveal their identity: Yigang Xu (Reviewer #2)

Transaction Report:

DOI: <https://doi.org/10.1128/spectrum.00309-24>

Re: Spectrum00309-24 (Integrative RNA-seq and ChIP-seq analysis unveils metabolic regulation as a conserved antiviral mechanism of chicken p53)

Dear Dr. Lu Cui:

Thank you for the privilege of reviewing your work. Below you will find my comments, instructions from the Spectrum editorial office, and the reviewer comments.

Revision Guidelines

Sincerely,
Manjula Kalia
Editor
Microbiology Spectrum

Reviewer #1 (Comments for the Author):

In the article by Lu Cui and colleagues, the author investigates the molecular networks related to chp53's antiviral activity against IBDV, ALV-J, and ILTV. ChIP-qPCR results indicate that chp53 regulates host cellular metabolism across various viruses.

The following are comments on manuscript

1. Please incorporate the multiplicity of infection (MOI) of viruses in the materials and methods section.

2. Figure 7: Please clarify why only metabolic genes were chosen for validation.
3. For Figure 7, please revise the figure legends to include the significance of the data and the number of experiments conducted.
4. The authors should elaborate on how the identified pathways are targeted for therapeutic purposes in the context of viral infection.

Reviewer #2 (Comments for the Author):

The manuscript by Lu et al. presents a comprehensive study on the antiviral role of p53, a well-established antiviral factor in numerous human and animal viruses, including chicken viruses. The authors have performed an integrative analysis of RNA-seq and ChIP-seq data across three major chicken pathogens: IBDV, ALV-J, and ILTV. Their findings suggest that targeting the virus-host metabolic interaction through the regulation of p53 could serve as a universal strategy for antiviral therapies in chickens. Overall, the manuscript exhibits a well-organized structure, introduces novel findings, and aligns its conclusions with existing results and relevant literature. Some minor improvements listed below are expected.

1. The introduction could benefit from a broader context on the impact of viral infectious diseases on humans and animals, as well as the transmission of pathogens from animals to humans. This would provide a comprehensive overview of the significance of the study.
2. Figure 8 appears to be of low quality, as if it was copied and pasted. It is recommended that the figure be edited to improve its clarity and presentation.
3. The manuscript explores the universal role of p53 in different avian viruses by jointly analyzing RNA-seq and ChIP-seq data. It is noted that the ChIP data is sourced from LMH cells, while the subsequent ALV-J transcriptome data is from DF-1 cells, and the data for IBDV is from bursal cells and IEL-NK cells. The potential impact of these differences on the analysis should be addressed in the Discussion section.
4. In Figure 7B, it is advisable to standardize the font to ensure consistency and avoid the use of both bold and non-bold fonts.
5. The manuscript should clearly state the number of experimental replicates performed in each experiment.
6. In the Conclusion section, it would be advantageous to underscore the significance of your research findings. Emphasizing how these insights contribute to our understanding of viral infection mechanisms and the development of antiviral strategies would highlight the impact of your work.

In the article by Lu Cui and colleagues, the author investigates the molecular networks related to chp53's antiviral activity against IBDV, ALV-J, and ILTV. CHIP-qPCR results indicate that chp53 regulates host cellular metabolism across various viruses.

The following are comments on manuscript

1. Please incorporate the multiplicity of infection (MOI) of viruses in the materials and methods section.
2. Figure 7: Please clarify why only metabolic genes were chosen for validation.
3. For Figure 7, please revise the figure legends to include the significance of the data and the number of experiments conducted.
4. The authors should elaborate on how the identified pathways are targeted for therapeutic purposes in the context of viral infection.

*Accepted

The manuscript by Lu et al. presents a comprehensive study on the antiviral role of p53, a well-established antiviral factor in numerous human and animal viruses, including chicken viruses. The authors have performed an integrative analysis of RNA-seq and ChIP-seq data across three major chicken pathogens: IBDV, ALV-J, and ILTV. Their findings suggest that targeting the virus-host metabolic interaction through the regulation of p53 could serve as a universal strategy for antiviral therapies in chickens. Overall, the manuscript exhibits a well-organized structure, introduces novel findings, and aligns its conclusions with existing results and relevant literature. Some minor improvements listed below are expected before the manuscript is formally published

1. The introduction could benefit from a broader context on the impact of viral infectious diseases on humans and animals, as well as the transmission of pathogens from animals to humans. This would provide a comprehensive overview of the significance of the study.
2. Figure 8 appears to be of low quality, as if it was copied and pasted. It is recommended that the figure be edited to improve its clarity and presentation.
3. The manuscript explores the universal role of p53 in different avian viruses by jointly analyzing RNA-seq and ChIP-seq data. It is noted that the ChIP data is sourced from LMH cells, while the subsequent ALV-J transcriptome data is from DF-1 cells, and the data for IBDV is from bursal cells and IEL-NK cells. The potential impact of these differences on the analysis should be addressed in the Discussion section.
4. In Figure 7B, it is advisable to standardize the font to ensure consistency and avoid the use of both bold and non-bold fonts.
5. The manuscript should clearly state the number of experimental replicates performed in each experiment.
6. In the Conclusion section, it would be advantageous to underscore the significance of your research findings. Emphasizing how these insights contribute to our understanding of viral infection mechanisms and the development of antiviral strategies would highlight the impact of your work.

Dear Dr. Manjula Kalia,

Thank you for your letter dated April 23rd, 2024! We would like to express our gratitude to you and the reviewers for dedicating your time and effort to reviewing our paper, which is crucial for improving the quality of our manuscript. We have thoroughly considered all comments and suggestions, and accordingly revised the introduction, discussion, and other sections. We also proofread the manuscript to minimize typographical and grammatical errors, and the revised manuscript has been provided as an attachment.

Here below is our description on revision according to the reviewers' comments. All modifications have been highlighted with yellow background in the revised manuscript.

Reviewer #1 (Comments for the Author):

In the article by Lu Cui and colleagues, the author investigates the molecular networks related to chp53's antiviral activity against IBDV, ALV-J, and ILTV. ChIP-qPCR results indicate that chp53 regulates host cellular metabolism across various viruses.

The authors' answer:

Dear Reviewer,

Thank you very much for your time and effort spent on reviewing our manuscript! Your insights are invaluable in enhancing the quality of our work. We have addressed each of your comments and suggestions in our revised manuscript, where all modifications have been highlighted for your convenience. Here below is our detailed point-by-point description on revision according to your comments and suggestions. We appreciate your attention to detail and your contribution to improving our study.

The reviewer's comment 1:

Please incorporate the multiplicity of infection (MOI) of viruses in the materials and methods section.

The authors' answer:

Thank you very much for your suggestion! We apologize for the omission of details such as the multiplicity of infection! In the revised manuscript, we have added information regarding the multiplicity of infection and the sample collection time in the Materials and Methods section (page 6, line 113-115).

The reviewer's comment 2:

Figure 7: Please clarify why only metabolic genes were chosen for validation.

The authors' answer:

We sincerely apologize for any confusion caused by our unclear description! The reason we chose to validate only metabolic genes in Figure 7 is because our integrated analysis of ChIP-seq and RNA-seq data revealed that metabolic pathways are commonly regulated by chp53 in chicken cells infected with ILTV, ALV-J, and IBDV. Therefore, seven of these genes were detected to verify whether these conserved metabolic genes are direct targets of chp53. Related explanation has been added in the Result section of the revised manuscript (page 14, line 293-296). We appreciate your attention to this detail and hope this clarification addresses your concern.

The reviewer's comment 3:

For Figure 7, please revise the figure legends to include the significance of the data and the number of experiments conducted.

The authors' answer:

Thank you for your thoughtful reading and valuable suggestions! We sincerely apologize for any confusion that may have arisen from the omission of crucial information in our initial description. We have now clarified the number of biological replicates for each experiment and the significance of the data in the legend of the revised manuscript (page 30, line 643-644). Additionally, information regarding the significance analysis has been annotated in Figure 7.

The reviewer's comment 4:

The authors should elaborate on how the identified pathways are targeted for therapeutic purposes in the context of viral infection.

The authors' answer:

Thank you very much for your suggestions, which have contributed significantly to the improvement of our article! Following your advice, two sections have been added to the Discussion of the revised manuscript, specifically addressing the impact of metabolic pathway alterations on viral replication during viral infection, and the therapeutic use of cellular metabolic pathways in treating viral infections. Following additional references have been added in the revised manuscript (page 18, line 386-400; page 19, lines 402-404).

Thank you once again for your time and effort spent on reviewing our manuscript!

Your expertise and insights greatly contribute to the quality of our work.

References:

1. Liu J, Zhang C, Hu W, Feng Z. 2019. Tumor suppressor p53 and metabolism. *J Mol Cell Biol* 11:284-292.
2. Xia W, Jiang P. 2024. p53 promotes antiviral innate immunity by driving hexosamine metabolism. *Cell Rep* 43:113724.
3. Bensaad K, Tsuruta A, Selak MA, Vidal MN, Nakano K, Bartrons R, Gottlieb E, Vousden KH. 2006. TIGAR, a p53-inducible regulator of glycolysis and apoptosis. *Cell* 126:107-20.
4. Li J, Wang Y, Deng H, Li S, Qiu HJ. 2023. Cellular metabolism hijacked by viruses for immunoevasion: potential antiviral targets. *Front Immunol* 14:1228811.
5. Matoba S, Kang JG, Patino WD, Wragg A, Boehm M, Gavrilova O, Hurley PJ, Bunz F, Hwang PM. 2006. p53 regulates mitochondrial respiration. *Science* 312:1650-3.
6. Rajendra R, Malegaonkar D, Pungaliya P, Marshall H, Rasheed Z, Brownell J, Liu LF, Lutzker S, Saleem A, Rubin EH. 2004. Topors functions as an E3 ubiquitin ligase with specific E2 enzymes and ubiquitinates p53. *J Biol Chem* 279:36440-4.
7. Maddocks OD, Vousden KH. 2011. Metabolic regulation by p53. *J Mol Med (Berl)* 89:237-45.
8. Hirabara SM, Gorjao R, Levada-Pires AC, Masi LN, Hatanaka E, Cury-Boaventura MF, da Silva EB, Santos-Oliveira LCD, Sousa Diniz VL, Serdan TAD, de Oliveira VAB, de Souza DR, Gritte RB, Souza-Siqueira T, Zambonato RF, Pithon-Curi TC, Bazotte RB, Newsholme P, Curi R. 2021. Host cell glutamine metabolism as a potential antiviral target. *Clin Sci (Lond)* 135:305-325.

Reviewer #2 (Comments for the Author):

The manuscript by Lu et al. presents a comprehensive study on the antiviral role of p53, a well-established antiviral factor in numerous human and animal viruses, including chicken viruses. The authors have performed an integrative analysis of RNA-seq and ChIP-seq data across three major chicken pathogens: IBDV, ALV-J, and ILTV. Their findings suggest that targeting the virus-host metabolic interaction through the regulation of p53 could serve as a universal strategy for antiviral therapies in chickens. Overall, the manuscript exhibits a well-organized structure, introduces novel findings, and aligns its conclusions with existing results and relevant literature. Some minor improvements listed below are expected.

The authors' answer:

Dear Reviewer,

Thank you very much for dedicating your valuable time and effort to reviewing our manuscript! Your insights are crucial in improving the quality of our work. We have addressed each of your comments and suggestions in the revised manuscript, highlighting all modifications for your convenience. We appreciate your attention to detail and your contribution to enhancing our research. Below is a detailed point-by-point description of the revisions made in response to your comments and suggestions.

The reviewer's comment 1:

The introduction could benefit from a broader context on the impact of viral infectious diseases on humans and animals, as well as the transmission of pathogens from animals to humans. This would provide a comprehensive overview of the significance of the study.

The authors' answer:

Thank you for your valuable suggestions! we have incorporated your advice by adding related descriptions in the Introduction section of revised manuscript (page 4, line 74-79). Following additional references have been added in the revised manuscript.

References:

1. Lloyd-Smith JO, George D, Pepin KM, Pitzer VE, Pulliam JR, Dobson AP, Hudson PJ, Grenfell BT. 2009. Epidemic dynamics at the human-animal interface. *Science* 326(5958):1362-1367.
2. Mollentze N, Streicker DG. 2020. Viral zoonotic risk is homogenous among taxonomic orders of mammalian and avian reservoir hosts. *Proc Natl Acad Sci USA* 117(17):9423-9430.

The reviewer's comment 2:

Figure 8 appears to be of low quality, as if it was copied and pasted. It is recommended that the figure be edited to improve its clarity and presentation.

The authors' answer:

We apologize for the low resolution of this figure! Following your advice, we have updated all images, including Figure 8, with high-resolution ones in the revised manuscript.

The reviewer's comment 3:

The manuscript explores the universal role of p53 in different avian viruses by jointly analyzing RNA-seq and ChIP-seq data. It is noted that the ChIP data is sourced from LMH cells, while the subsequent ALV-J transcriptome data is from DF-1 cells, and the data for IBDV is from bursal cells and IEL-NK cells. The potential impact of these differences on the analysis should be addressed in the Discussion section.

The authors' answer:

Thank you very much for your thorough review of our manuscript and valuable suggestions! The potential impact of different cell sources on our analysis has been commented in the revised Discussion section according to your suggestion (page 17, lines 366-369).

The reviewer's comment 4:

In Figure 7B, it is advisable to standardize the font to ensure consistency and avoid the use of both bold and non-bold fonts.

The authors' answer:

Thank you for your careful reading and suggestions! Following your advice, we have adjusted the font in Figure 7 as per the revised manuscript.

The reviewer's comment 5:

The manuscript should clearly state the number of experimental replicates performed in each experiment.

The authors' answer:

Thank you for your careful reading and suggestions! We apologize for any confusion caused by the lack of description regarding biological replicates in our initial manuscript. We have now clarified the number of biological replicates and the

significance of the data in the legend of the revised manuscript (page 30, line 643-644). Additionally, information regarding the significance analysis has been annotated in Figure 7.

The reviewer's comment 6:

In the Conclusion section, it would be advantageous to underscore the significance of your research findings. Emphasizing how these insights contribute to our understanding of viral infection mechanisms and the development of antiviral strategies would highlight the impact of your work.

The authors' answer:

Thank you very much for your suggestion, which has made a valuable contribution to the improvement of our article! We have incorporated two key aspects into the Discussion section based on your advice: firstly, emphasizing the significant role of p53 in antiviral processes through regulating cellular metabolic pathways, and secondly, highlighting the importance of modulating metabolic pathways as a crucial strategy for treating viral infections (page 18, lines 386-400; page 19, lines 402-404). These additions aim to underscore the importance of our research in advancing the understanding of viral infection mechanisms and the development of novel antiviral strategies. Following additional references have been added in the revised manuscript.

Thank you once again for your great effort on our manuscript, which is valuable in improving the quality of our manuscript!

References:

1. Liu J, Zhang C, Hu W, Feng Z. 2019. Tumor suppressor p53 and metabolism. *J Mol Cell Biol* 11:284-292.
2. Xia W, Jiang P. 2024. p53 promotes antiviral innate immunity by driving hexosamine metabolism. *Cell Rep* 43:113724.
3. Bensaad K, Tsuruta A, Selak MA, Vidal MN, Nakano K, Bartrons R, Gottlieb E, Vousden KH. 2006. TIGAR, a p53-inducible regulator of glycolysis and apoptosis. *Cell* 126:107-20.
4. Li J, Wang Y, Deng H, Li S, Qiu HJ. 2023. Cellular metabolism hijacked by viruses for immunoevasion: potential antiviral targets. *Front Immunol* 14:1228811.
5. Matoba S, Kang JG, Patino WD, Wragg A, Boehm M, Gavrilova O, Hurley PJ, Bunz F, Hwang PM. 2006. p53 regulates mitochondrial respiration. *Science* 312:1650-3.
6. Rajendra R, Malegaonkar D, Pungaliya P, Marshall H, Rasheed Z, Brownell J, Liu LF, Lutzker S, Saleem A, Rubin EH. 2004. Topors functions as an E3 ubiquitin ligase with specific E2

- enzymes and ubiquitinates p53. *J Biol Chem* 279:36440-4.
7. Maddocks OD, Vousden KH. 2011. Metabolic regulation by p53. *J Mol Med (Berl)* 89:237-45.
 8. Hirabara SM, Gorjao R, Levada-Pires AC, Masi LN, Hatanaka E, Cury-Boaventura MF, da Silva EB, Santos-Oliveira LCD, Sousa Diniz VL, Serdan TAD, de Oliveira VAB, de Souza DR, Gritte RB, Souza-Siqueira T, Zambonato RF, Pithon-Curi TC, Bazotte RB, Newsholme P, Curi R. 2021. Host cell glutamine metabolism as a potential antiviral target. *Clin Sci (Lond)* 135:305-325.

Re: Spectrum00309-24R1 (Integrative RNA-seq and ChIP-seq analysis unveils metabolic regulation as a conserved antiviral mechanism of chicken p53)

Dear Prof. Hai Li:

Your manuscript has been accepted, and I am forwarding it to the ASM production staff for publication. Your paper will first be checked to make sure all elements meet the technical requirements. ASM staff will contact you if anything needs to be revised before copyediting and production can begin. Otherwise, you will be notified when your proofs are ready to be viewed.

Sincerely,
Manjula Kalia
Editor
Microbiology Spectrum